# Second Time University Program as a Public Policy: Contributions and Limitations to Physical Leisure Activities and Health Promotion

**DOI:** 10.3390/sports13070207

**Published:** 2025-06-26

**Authors:** Alex Caiçara de Albuquerque, Junior Vagner Pereira da Silva

**Affiliations:** Faculty of Medicine, Federal University of Mato Grosso do Sul 1, Campo Grande 79070-900, Brazil; albuquerquealex27@gmail.com

**Keywords:** higher education, situational barriers, public policies, leisure physical activities, health promotion

## Abstract

The Second Time University Program is a federal government policy instituted in 2009. Given its importance, this study aimed to analyze the program’s contributions and limitations in promoting leisure-time physical activity and student health. The study is a retrospective longitudinal, qualitative–quantitative, exploratory and documentary study, analyzing the guidelines and public notices between 2009 and 2023. The program is predominantly focused on integral development, with the inclusion of objectives aimed at health promotion, relating leisure-time physical activities to a physically active lifestyle and a reduction in sedentary lifestyles in 2020. Its scope is low and selective, with a presence in only 47.82% of universities. Continuity is also low, with the majority of universities only being covered by the program in one call for proposals. In conclusion, although it promotes leisure-time physical activity and holistic health promotion, the public policy is limited and discontinuous.

## 1. Introduction

The literature has shown that the majority of university students are physically inactive [1,2]. This situation has generated concern from different theoretical approaches, whether related to disease prevention or health promotion.

Based on the biological view of the health–disease process, disease prevention focuses on diseases arising from the prevalence of physical inactivity [3]. Following the same line of thought, the biomedical view of health promotion, aimed at reducing morbidity and mortality indicators for chronic degenerative diseases, adopts a conservative approach [4]. This approach is based on the idea that disease prevention and health promotion depend exclusively on people’s individual will and effort. This perspective disregards contextual aspects and requires changes in people’s lifestyles and habits [3], whether at the family or community level [5].

According to the conservative view, physical activity is understood as any bodily movement that requires energy expenditure above the resting level [6], whether at work, commuting, household chores or leisure [7]. In general, physical activities, especially leisure activities, are seen as direct and relevant mechanisms for preventing cardiovascular risks [2] and various other chronic degenerative diseases [4].

This theory hides the role of cultural, historical and economic issues, suggesting that individuals who perform strenuous work activities have a satisfactory state of health. In this way, the historically produced contradictions between what is obligatory and what is an option are disregarded, the latter being marked above all by desire, pleasure, happiness and, above all, opportunities, which in Brazil, as Knuth and Antunes [4] point out, are privileges that only a few have access to.

This approach has been constantly criticized because it is based on blaming the victim; i.e., it assumes that lifestyle is a personal choice. As Knuth and Antunes [4] point out, the behavioral approach to health promotion, based on the hegemonic pillars, assumes that simply changing lifestyle would be enough to establish a causal relationship between health and disease.

In the opposite direction is the critical approach, which seeks explanations based on cultural, historical and economic issues. The socio-environmental approach, defined as a set of strategies aimed at articulating with other public policies, aims to promote health through a dynamic construction related to socially and historically situated contexts. The aim is to improve living conditions and satisfy social health needs.

In this context, health promotion demands the mobilization of political, human and financial resources from different sectors, requiring intersectorality and the participation of the state and civil society [8]. As the authors point out,

“Health promotion is a broad concept; it goes beyond disease prevention actions and enables individuals to exercise their autonomy and achieve better living conditions. It is understood as the process that enables the population to exercise and increase control over their health, aiming for a state of well-being”.[8]

In this sense, it consists of a set of systematic, continuous and articulated actions that take into account the subject’s biological, social, affective, cognitive and cultural factors. Therefore, its design is based on understanding the structural, community and individual effects. As highlighted by Carvalho [9], health promotion through leisure physical activity (LPA) is recognized as important, presenting possibilities that go beyond the utilitarian and individualistic effects of protection against chronic non-communicable diseases.

Although biophysiological effects are recognized as influential factors, it is important not to limit studies to them, as involvement with LPA can also provide well-being and relationships [10], alleviate anxiety and depression [11,12], improve the state of well-being [13], expand social relationships [14] and improve quality of life [15].

Therefore, the importance of LPA should not only focus on justifying disease prevention or promoting biophysiological health. In addition to acknowledging these effects, it is essential to engage in such experiences, as interpersonal interactions and political and cultural developments occur through the use of available time in social settings such as sports parks, sports facilities, multi-purpose courts, gyms, clubs, schools, universities, among others.

When considering health promotion from a critical perspective (socio-environmental approach) and LPA based on the holistic model, it is understood that various factors act as barriers, preventing or hindering people from taking up their available time with LPA. In other words, the experiences in question, as opposed to what the behavioral approach proposes, are conditioned by factors that integrate four dimensions of barriers—environmental, social, physical and behavioral [16].

Among the sociocultural barriers, lack of time stands out, as observed in state university students in Bahia [17]. This limitation stems from involvement with studies, a long academic day, household chores [18] or a long working day [19].

In an LPA project promoted in the context of the university itself, the clash between undergraduate schedules and the modalities offered and the consumption of social time was pointed out as the main barrier to staying and possibly being responsible for future dropouts [20]. A similar result was observed in Chilean [21], Spanish [22] and Colombian [23] students.

The influence of lack of time on LPA opportunities by university students tends to increase as the semester system progresses [17] and during specific periods, such as assessment periods, especially among students who combine academic duties with domestic duties or employment [24]. In Brazil, among students in the final periods of their degree, lack of time has a greater influence due to the search for a job, participation in internships and course completion projects [17]. A similar situation was found in Chile, where first-year students had a more significant involvement, with a gradual decline as they progressed through the course [25].

Considering the holistic benefits (biophysiological, psychological, social, affective, cognitive and political) resulting from the experience of LPA, it is essential that involvement with it is encouraged and materialized in the university context. To this end, public policies need to be implemented with the aim of mitigating the problem and contributing to a favorable scenario for health promotion, since involvement with them is related to socio-environmental conditions.

Since involvement with LPA and health promotion is conditioned by socio-environmental factors, Silva and Reverdito [20] believe that free programs and the conditions of the university context are favorable mechanisms for students to occupy their time (when they have it) with this type of experience.

Public policies, conceived as resolutions to problems that have attracted the attention of public political agents through their incorporation into the agenda-setting, are then forwarded to experts in specific sectors to formulate solutions to the issues raised, so that visible actors, such as those elected by the people, can make their decisions on the alternative to be adopted to solve the problem [26]. Within the public administration, the government, as the leader of the federal, state, District and municipal Executive, is primarily responsible for the decision-making process.

Public policies aim to promote well-being, and their bias is linked to social change, income distribution and social equity [27]. To this end, the Executive Branch draws up plans (strategic macro-directives), which are implemented by programs of a tactical–managerial nature and operationalized by projects of an operational nature. According to Secchi and Machado [28], these documents are fundamental for defining the public policy instrument to be implemented, since they establish the objectives, targets, target audience and monitoring and evaluation mechanisms.

Considering the factors that affect the decrease in university students’ social time and the effects of its incorporation into people’s daily social lives, since it is a historically constructed experience, public policies aimed at LPA at university become relevant and urgent. According to Silva [29], “[…] this is a population in ecological transition—adolescence to adulthood and secondary school to higher education—with considerable contextual changes and cultural expectations” (p. 3).

The implementation of public policies aimed at LPA and health promotion in the university context has shown positive effects, as evidenced in the Brazilian scenario. The fact that services are free of charge and offered within the higher education institution itself have been identified as facilitators [20]. In addition, a sports program implemented in the context of a Brazilian federal university showed potential to contribute to an increase in LPA in 54.05% of participants [30].

This issue stems from the fact that involvement in this type of experience is not inherent to personal choices, but rather to the conditions created by public policies, including environmental ones, such as the existence of spaces and their attractiveness, programs with guidance from professionals, among others [31]. The creation of a favorable environment for LPA in the university context has corroborated greater involvement with such experiences, resulting in greater satisfaction with life [32], satisfaction with the institution, greater academic and social integration [33] and improvements in social skills among students [34]. In short, this environment has contributed to promoting the health of university students.

A preliminary theoretical analysis showed that, at the national level, there is a specific program aimed at LPA in higher education: the Second Time University Program. This is the only public policy created by the federal government to promote university sport in Brazil. The initiative in question is an extension of the Second Time Standard Program, developed by the Ministry of Sport’s National Secretariat for Amateur Sport, Education, Leisure and Social Inclusion. The program is organized at different levels, with the aim of ensuring development and implementation in higher education institutions through Decentralized Execution Terms or Agreements.

Considering that this is the only federal program specifically aimed at LPA in force in Brazil and that few studies have looked at the Second University Time Program as a public policy [19,20,35,36], but none of them related to health promotion, this study aims to analyze the Second University Time Program and its contributions limitations for the promotion of LPA and health promotion. Specifically, we sought to identify the objectives of the program and establish the relationship between LPA and health promotion, as well as determine the institutions covered and the scope and continuity of the program.

Public policy analysis contributes to the generation of information, arguments and consensus, giving greater solidity to public decisions through an analytical approach. In this way, superficial recommendations are avoided, and a professional analysis is approached, increasing the likelihood of success (26, 27). This segment of political science allows

[…] understanding the modes and general rules of operation of public action and analyzing their continuities and ruptures, as well as the processes and determinants of their development, and identifying the multiplicity of factors and forces that form the real processes of public policies.([37], p. 12)

The purpose of public policy analysis is to deconstruct consolidated understandings and produce alternatives (creative), foster arguments for political debate, mediate and resolve conflicts (argumentative) and legitimize alternatives by means of implementation (legitimizing) [26].

## 2. Materials and Methods

The study is characterized as a longitudinal retrospective, as it analyzes the phenomenon over a long period of time, referring to the past. As for the type of information, it is qualitative in nature. In terms of objectives, the study is classified as exploratory, i.e., a process of developing or discovering ideas and intuitions [38]. It is also classified as documentary, which consists of the analysis of electronic public files, from administrative publications and sources internal to the institution [39], which have not been analyzed, and which enables objective knowledge of reality [38].

The research technique was structured in five stages, as described by Gil [38]: formulation of the problem; preparation of the work plan; identification of sources; location of sources of material; analysis and interpretation.

The problem was formulated during the design phase of the first author’s dissertation, which established the guiding assumptions and then the schedule of actions to be carried out through the work plan. The work plan was conceived and developed in accordance with the project schedule, which was segmented into sequential stages: bibliographic survey, literature review, data collection, analysis and interpretation of results, publication of articles.

The identification and location of the sources, as well as the obtaining of the material, were the guidelines of the Second Time University Program, as well as the opening and results notices. The time frame selected for analysis covered the period from 2009 to 2023. The documents were accessed via the Ministry of Sport’s institutional website. Given the nature of the study, which covers 14 years of a national policy, and considering that, during the period in question, Brazil went through six presidential terms, access to the data proved to be complex, as it was not available on a single website. This indicates inadequate governance in the federal sphere of sport.

The data were analyzed and interpreted based on the analytical procedures proposed by Gil [38], which consist of defining the objectives or hypotheses, setting up a reference framework, selecting the documents to be analyzed and defining the units of analysis.

The objectives or hypotheses were defined during the exploratory phase, with the aim of bringing the researcher closer to the area and the object of investigation. The creation of a frame of reference helped to guide the research and interpret the data. The reference framework adopted was in line with the themes of public policies [26], physical activity [40], LPA [36], health promotion [4,7,41].

After selecting the documents on the Ministry of Sport’s website, they were all transferred to an online database on Drive, organized by year and then opened in Adobe software.

The units of analysis established and selected for the study focus on the general and specific objectives, the institutions covered and the scope and continuity of the program.

The qualitative research was based on content analysis, which consists of a process of inference based on the preliminary reading of documents. Specifically, the co-occurrence analysis technique was adopted, which seeks to identify, in a text, the relationships between the parts of a message and the possible similarity between two elements. To do this, it was necessary to follow these steps: first, select the recording unit (expression) and the categories of themes (LPA and health promotion); then, choose the context units (objectives of the Second University Time Program in the different versions of the guidelines); then, identify the presence or absence of each recording unit in the context units; and finally, analyze and interpret the data [42].

The data were interpreted using the creative and argumentative public policy analysis method, as recommended by Secchi [26].

## 3. Results

Over the 16 years of the University Second Time Program’s existence, the program has undergone different changes in terms of its objectives. In 2009 and 2016, the general objective began to focus on the creation of university centers and the reframing of sport as a manifestation of body culture. Between 2020 and 2021, the concept of corporal practices through quality educational sport was implemented. From 2023 onwards, changes were made to the scope of the program, which was no longer limited to federal higher education institutions, but also included state and municipal higher education institutions (Table 1).

With regard to the specific objectives, the program in 2009 was focused on educational sport, recreation and leisure, linked to the production of knowledge in the field of sport, participatory management and integration of the university community, as well as the involvement of academic associations. In addition, the aim was to rescue and value body culture, include sport in the university’s pedagogical project and support researchers and scientific organizations. In 2016, in addition to the specific objectives set in 2009, the aim was to guarantee a variety of sports, raise awareness among those who do not practice sport, encourage actions based on the university tripod, provide internships for students in the area and recognize training sport as a possibility. Between 2020 and 2021, the specific objectives were summarized as maintaining an active life through sport and the all-around development of participants, the social and cultural appreciation of bodily practices and the integration of educational sports policies with other areas. From 2023, the specific objectives were expanded to include sports training as a complementary alternative and tackling sedentary lifestyles among young people and adults.

With regard to health promotion, however, there is no explicit expression of this intention in the program. However, in an implicit way, aspects that bring it closer to a holistic view of LPA and socio-environmental health promotion be seen, since the 2020, 2021 and 2023–2026 guidelines presented objectives aimed at integral development and encouraging university students to maintain a physically active life, something that is repeated in 2021 and 2023. The results are presented descriptively (Table 1) and by frequency (Table 2).

Over the course of 16 years, only 33 of Brazil’s 69 federal universities (47.82%) have been awarded at least once with a nucleus of the Second University Time Program (Table 2). Despite an increase in the number of proposals approved in 2023, indicating progress in LPA public policies in the university context, none of the 17 approved proposals were actually implemented. The federal government’s justification for not decentralizing funds was the financial crisis, which prevented universities from purchasing sports equipment and hiring human resources, such as teachers and monitors.

Regarding the continuity of the centers, this was even lower, since 23 (61.69%) of the 33 universities that received funding did so only once. Furthermore, no university received funding in all the calls for proposals.

Of the universities that received funding, seven are located in the Central-West region. The University of Brasilia (UnB) and the Federal University of Mato Grosso do Sul (UFMS) are the institutions in the Midwest, as well as in Brazil as a whole, that have received the most grants under the program.

## 4. Discussion

Health promotion through LPA is not an explicitly stated objective in the program’s guidelines, which are more geared towards student development through sport as a participatory and educational experience. It was not until 2020 that specific objectives were added aimed at holistic development and encouraging university students to maintain an active lifestyle associated with the practice of sport. In 2021, these objectives were maintained, but in 2023, they were replaced by the fight against sedentary lifestyles.

The absence of explicit objectives for health promotion until 2019 and the implicit attention to this issue from 2020 onwards leads us to some reflections. The first is the absence of explicit objectives aimed at health promotion and its inclusion, even if implicitly and indirectly. Until 2020, this indicates the belated intentionality of the Second Time University Program in relation to public policy aimed at health promotion.

The need for a commitment from all sectors to issues related to health promotion dates back to 1993, with the Hensiki Declaration [43]. In Brazil, since 2002, documents related to the National Health Promotion Policy have reinforced the relevance of actions, programs and policies taking place within (intrasectorality) and outside the sector (intersectorality) responsible for health [44].

Furthermore, a second issue is that, in addition to health promotion being included late in the program in 2020, it was based on active living and sedentary lifestyles, i.e., it was linked to lifestyle changes. However, it must be considered that the implementation of public policies aimed at democratizing access is a relevant social aspect. This factor is relevant because it is not based on blaming the victim or understanding that the university student’s involvement with LPA is a choice, thus moving away from the conservative side of health promotion. The conservative approach is based on prescribing and providing decontextualized information [3], without considering the impact that environmental aspects have on involvement with such experiences [31].

Behavioral changes may be necessary and are supported by the critical approach, and the issue to be considered is the relationship among subjects, workers and services and the way in which public policies act to change contexts, in order to favor involvement with LPA experiences, that is, the construction of possibilities [3]. Therefore, for changes to occur, it is essential that public policies are developed and cover urban infrastructure, street connectivity, access to parks, trails, active transportation and urban zoning [31].

The creation of public policies aimed at expanding leisure opportunities for students is essential for health promotion, because, contrary to the conservative view, involvement in LPA is not a choice but a privilege granted to a few. In this sense, public policies, in the light of social justice, should be developed with a view to reducing existing discrepancies [4].

In this sense, there is agreement with Castiel [45], who recognizes the positive effects on the health of people who achieve behavioral changes. However, it is necessary to recognize that the public policies in question have limitations in the face of a complex issue related to socio-economic disparities. Therefore, they distance themselves from the conservative view of health promotion, which culminates in blaming the individual. In this sense, as Bagrichevsky and Estevão [41] point out, it is imperative to consider the relationship between the subject and culture, as well as the interference of government agendas.

Health promotion is a complex action that requires an understanding of the connections among the economic, political and cultural events to which people are exposed in their daily lives [46]. When considering the Second Time University Program as a public policy that could contribute to expanding access to LPA and promoting health, this consideration is not made naively, as if it were enough. It is important to note that such measures are palliative and that elements related to social barriers, such as lack of time, sometimes condition and determine such experiences.

Therefore, public policies on university experience should offer projects in the university environment itself, so that students can take advantage of this service. However, even though activities are offered in this context, other factors can prevent a significant proportion of students from taking advantage of these activities. As the literature has shown, lack of time due to concomitant studies, work, household chores [18] and long working hours [19] are factors responsible for non-participation, which is exacerbated during assessment periods [24] and semester progression [47].

On certain occasions, the simultaneity of activities experienced by students can result in limited time for social experiences, even during breaks, which are used for extracurricular activities [48]. This position is pertinent, insofar as non-involvement in LPA should not be a reason to blame people but rather to reflect on historically produced social inequalities.

This is because, although LPA in the university context favors the incorporation of these experiences into everyday life, we cannot lose sight of the impacts of other conditioning factors, such as those arising from social contradictions, especially those related to people who study, work, live far from the university and use public transport.

However, when we consider integration as one of the objectives, we see a broader concept of health promotion and holistic LPA, since it covers biophysiological, social, political and cultural aspects. From this broader perspective, individuals who practice LPA show improvements that go beyond the biophysiological aspects, as there are also improvements in psychological health, stress control [10], a reduction in anxiety and depression [12,48], quality of life and the strengthening of social ties [14]. Recent studies show that university students who practice LPA tend to have better self-esteem [49] and quality of life [50].

LPAs can be understood as holistic experiences, which according to Piggin [40], operate in multiple spaces and contexts, reflected in interactions, emotions, ideas, instructions and social ties. LPA, as advocated by Carvalho, Cohen and Akerman [3] for physical activities and bodily practices, must be analyzed in a complex context, in which individual behavior is related to social, cultural, material and economic aspects.

The creation of a program focused primarily on serving university students is an important initiative in the context of Brazilian public policy. Since 1996, as evidenced by Silva [20], with the elimination of mandatory physical education in higher education, students have been neglected by federal public policies focused on LPA. Although the National Student Assistance Program (NSAP) was conceived in 2008, preceding the Second Time University Program, it was designed with the objective of democratizing the conditions for remaining in higher education, minimizing the effects of social inequalities and reducing retention and dropout rates, primarily among students in situations of socioeconomic vulnerability.

Considering that sport is included among the priority areas for the application of NSAP resources, given the existing system of prioritization within Brazilian public policies [51], sport is still secondary. According to Fava and Cintra [52], the program’s resources were predominantly allocated to housing, food, transportation, daycare and educational support among the 69 Brazilian federal universities. Januário et al. [53] observed that the resources were allocated to food, educational support, health care and digital inclusion. This results in a low participation of sports activities financed with program resources [53,54,55], reaching 0.50% of the investment [55]. Given the discretion granted to university rectors, the use of these resources is uncertain, inconsistent and discontinuous, depending on the political will of the administrators.

Regarding the scope and continuity of the program, it was possible to see that this is low, either because it does not cover all federal universities, or because, among the covered, most only received it for one year, through a single call for proposals. This is a problem, as it only covered a few Federal Higher Education Institutions and reduced the possibilities for practicing LPA, directly reflecting on the opportunities for students to experience and get involved and, consequently, reducing the possibilities for the social context to create suitable conditions for university students to experience LPA in the context of their academic training.

This indicates that although the Second Time University Program may have promoted health in the university context, it did not have the characteristics and attention needed by the four presidents who governed Brazil between 2009 and 2023, to the point of transforming the program into a social policy. This is because, throughout its existence, the program was highly selective and had low coverage, as evidenced in this investigation. According to Bagrischevsky et al. [56], for a social policy to be successful in reducing inequalities, certain requirements must be met.

Furthermore, the low value placed on the program in the federal context was evident in the last call for proposals in 2023, as none of the 17 approved proposals were implemented due to the federal government not making funds available. This situation highlights the high selectivity and hierarchization of priorities within the federal agenda-setting [57], with public policies aimed at sport being sidelined [51]. These decisions are sometimes centered on the political will of governments or the pressure exerted by society [58].

In view of this, it is necessary to create possibilities to extend the program to new institutions and maintenance mechanisms, considering that it can provide new social, political and cultural experiences through access to different sports, which can result in holistic health promotion.

Specifically, with regard to the Midwest, the data suggest that the federal universities have a good capacity to raise funds made available by the Ministry of Sport for the implementation of the Second University Time Program, with the exception of the UFR, all the others having been contemplated. In addition, UnB and UFMS have managed to maintain a certain continuity in the development of the program, standing out on the national scene.

The low coverage of the program leads us to reflect on the fact that university students do not take up the time available for LPA experiences. Firstly, the time available is scarce in view of the academic profile coming from vulnerable socio-economic classes, as the literature shows. Secondly, there is the absence (or limited presence) of public policies aimed mitigating the effects of social contradictions on the population’s leisure opportunities. In this sense, despite its importance, it is necessary to recognize that, in a complex context, the Second Time University Program does not appear as a national policy capable of promoting health in a comprehensive way, showing signs of its extinction as a program aimed at LPA for university students, since, after the non-payment of the proposals approved in 2023, no other call for proposals was launched.

## 5. Conclusions

The Second Time University Program does not explicitly focus on health promotion but mainly on participation. However, since 2020, objectives have been included that aim to promote integral development, encourage a physically active lifestyle and combat sedentary lifestyles.

Based on this analysis, it can be understood that two of its objectives (promoting an active lifestyle and reducing sedentary lifestyles) are aimed at promoting health based on a holistic approach. Although aspects related to physically active lifestyles and sedentary lifestyles have historically been linked to conservative approaches, such as biomedical and behavioral ones, the Second Time University Program recognizes that these conditions are not the choice of the individual but the result of conditions and opportunities (or lack thereof). It is no coincidence that the program is presented as a political tool aimed at democratizing students’ access to LPA in the university context.

Contrary to the conservative approach, which blames the individual, the program recognizes the socio-economic, political and cultural effects on lifestyle, adopting a critical socio-environmental approach. In addition, it aims to promote the holistic development of university students and the possibility of adjusting the modalities offered to the local cultural reality, thus bringing the holistic approach to physical activity closer to the socio-environmental approach to health promotion.

Although it has the potential to expand LPA opportunities in the university context for students and promote health, the implementation of the program through public notices has proven to be low-reaching and not universal, given that, in 16 years of existence, it has not reached 50% of Brazilian federal universities, and most of those contemplated were only for one year.

The conclusion is that, although it has the potential to promote health from the holistic approach of physical activity and the socio-environmental approach to health promotion, the Second Time University Program has limited conditions to achieve these ends, given its limited and discontinuous reach, including the non-payment of approved proposals in 2023 and the absence of new calls for proposals since then, signaling the program’s inertia.

Finally, the authors acknowledge the limitations of the study, because although documentary analysis contributes to the production of knowledge about public policies, and this methodological resource should be encouraged, the technique imposes certain limitations, such as knowledge about how users perceive the program and the reasons why several federal universities never applied for the program or were contemplated. It is therefore recommended that future studies be carried out to deepen scientific production on the Second Time University Program.

## Figures and Tables

**Table 1 sports-13-00207-t001:** General and specific objectives of the Second Time University Program—Brazil, 2009/2023.

Objectives		
Year	General	Specific
2009	To democratize access to sports for the academic community of public universities, primarily the student body, by promoting the creation of university centers of the Second Time Program, as a way of enabling the reframing of sport as a manifestation of body culture.	✔ Offering educational sports practices, to meet the needs of sports training, recreation and leisure, rescuing and raising the body culture of the beneficiaries of the program; ✔ Diversify the range of activities on offer, valuing other bodily practices; ✔ To provide suitable conditions for quality educational sports practice, focusing on the all-around education of participants; ✔ Inserting sport as a cross-cutting action in the university’s pedagogical project; ✔ Encouraging the production of knowledge in the field of university sport by supporting researchers, institutions and scientific bodies; ✔ Get to know the students’ reality and interests better; ✔ Integrate into the university community; ✔ Develop participatory management; ✔ Develop sports activities to integrate the university community; ✔ Broaden the involvement of the academic guilds.
2011	No information	No information
2016	To democratize access to sports practice for the academic community of higher education institutions, primarily the student body, by offering the development of educational sports centers, offering the opportunity to broaden specific knowledge through the experience of concrete teaching—learning situations.	✔ Offering educational sports practices to meet the needs of sports training and development, recreation and leisure, rescuing and raising the body culture of the beneficiaries of the program, with performance sport being a possible alternative to be made available; ✔ Diversify the range of activities on offer by highlighting other body practices; ✔ To provide suitable conditions for quality educational sports practice, focusing on the all-around education of participants; ✔ Inserting sport as a transversal action in the university’s pedagogical project, offering internship opportunities to students in the field of physical education or sport; ✔ Develop sports activities to integrate the university community; ✔ Ensuring that different sports are on offer, as well as raising awareness among those who do not practice sport; ✔ Encourage coordination with teaching, research and extension activities.
2020	To democratize the access of the university community (of public higher education institutions), primarily the student body, to the contents of bodily practices through quality, educational sport.	✔ Encourage university students to maintain an active life linked to the practice of sport; ✔ Offer body practices that stimulate the all-around development of the participants; ✔ Stimulate the social and cultural values inherent in bodily practices; ✔ Motivate the promotion of inter-ministerial actions that integrate educational sports policy with other sectors (education, health, culture, defense, among others).
2021	To democratize the access of the university community (of public educational institutions), primarily the student body, to the contents of bodily practices through quality, educational sport.	✔ Encourage university students to maintain an active life linked to the practice of sport; ✔ Offer body practices that stimulate the all-around development of the participants; ✔ Stimulate the social and cultural values inherent in bodily practices; ✔ Motivate the promotion of inter-ministerial actions that integrate educational sports policy with other sectors (education, health, culture, defense, among others).
2023–2026	To democratize access to sports and physical activity for students at public higher education institutions (HEIs) and state and municipal public higher education institutions, offering the development of educational sports centers and/or sports training, with the aim of broadening specific knowledge of sports and leisure.	✔ Offering educational sports practices, to meet the needs of sports training and development, recreation and leisure, rescuing and raising the body culture of the beneficiaries of the program, with sports training being a possible alternative to be made available; ✔ Promote the offer of activities valuing diversified bodily practices; ✔ To offer suitable conditions for the practice of educational sport and/or quality sports training, focusing on the integral training of the participants; ✔ To include sport as a cross-cutting action in the pedagogical project of public higher education institutions (HEIs) and state and municipal public higher education institutions, offering internship opportunities to students in the field of physical education or sport; ✔ Develop sports activities to integrate the university community; ✔ Ensuring that different sports are on offer, as well as raising awareness among those who do not play sports; ✔ Encouraging links with teaching, research and extension activities through the production of knowledge and the practice of sport and leisure; ✔ To help tackle the high levels of sedentary lifestyles among our young people and adults; ✔ Encourage the inclusion in the National Sports Development Network of partner sports and leisure institutions and administrations; ✔ Contribute to tackling violence in higher education institutions by promoting a culture of peace through sport.

**Table 2 sports-13-00207-t002:** Federal universities with approved Second Time University Program centers—Brazil, 2009–2023.

University	Year of Approval						
	2009	2011	2012	2017	2018	2020	2023
Federal University of ABC		x					x
Federal University of Alagoas		x				x	
Federal University of Amazonas		x				x	
Federal University of Espírito Santo		x					
Federal University of Ceará		x					
Federal University of Goiás		x				x	x
Federal University of Jatai						x	x
Federal University of Juiz de Fora		x					x
Federal University of Lavras		x					
Federal University of Maranhão		x					
Federal University of Minas Gerais	x					x	
Federal University of Mato Grosso do Sul		x		x		x	x
Federal University of Mato Grosso						x	
Federal University of Ouro Preto	x						
Federal University of Pará		x					x
Federal University of Pernambuco		x					
Federal University of Paraná		x					
Federal University of Reconcavo Baiano		x					x
Federal University of Rio de Janeiro		x			x		
Federal Rural University of Rio de Janeiro			x				
Federal University of Rio Grande do Norte		x					
Federal Rural University of Pernambuco		x		x			
Federal University of Sergipe						x	x
Federal University of Santa Catarina						x	
Federal University of São Carlos						x	
Federal University of São João Del-Rei		x					
Federal University of Santa Maria	x	x				x	
Federal University of Univerlândia						x	
Federal University of Viçosa		x				x	x
Federal University of Brasília	x	x		x		x	
Federal University of Alfenas		x					
Federal University of Mampa		x					
Federal Technical University of Paraná		x					
Federal University of Oeste Baiano							x
Federal Institute Maranhão							x
Federal University of Piauí							x
Federal University of Rio Grande do Sul							x
Federal University of Catalão							x
Federal University of Grande Dourados							x
Federal University of Triângulo Mineiro							x
Federal University of Itajubá							x

Note: Prepared by the authors.

## Data Availability

The data is available at https://www.researchgate.net/publication/392267803 (accessed on 16 May 2025).

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
