# Peer review of "Second Time University Program as a Public Policy: Contributions and Limitations to Physical Leisure Activities and Health Promotion"

_sports, 2025, doi:10.3390/sports13070207_

Round 1

Reviewer 1 Report

Comments and Suggestions for Authors

Abstract:  Well-written and covers the content of the manuscript.

Keywords: Appropriate and relevant. 

Introduction and literature review: Good description of the different approaches to understanding how physical activity and health promotion may be considered (such as biological, critical).  Good development of the rationale for the research. It would be useful to provide a little more analysis of the role of public policy in influencing public choices.  What are some examples that have worked (e.g., anti-smoking campaigns), when has it not worked? What are the factors that have been most notable in campaigns with success?    

Some minor grammatical errors to consider include:

Broken sentence at line 58 "Furthermore, as the authors point."

Define or spell out the acronym for AFL at line 67 as this is the first time it is used.  

Materials and Methods:  Good consideration of the process that was used for the project.  It would be useful to include more details in the analysis section.  For example, what was learned during the various steps?  Where there any changes in decision making along the way?  What insights were obtained at each step? 

Results:  The use of the tables to summarize the results and changes in approach of the universities over the time period were a useful presentation of data. Did the researchers find any key differences over the time periods that particularly stood out as impactful to making change in students' practices?  The time period of 2023-2026 had the greatest number of activities being reported.  Did they seem to be on track for achieving these - given the time period has not been completed?  

Discussion:  The discussion section would benefit from further expansion.  The authors have outlined the key areas identified through their results and now can further expand upon them. For example, expand on the following paragraph as it is only one sentence: (line 34) "The absence of explicit objectives for health promotion until 2019 and the implicit attention to this issue from 2020 onwards lead us to analyze them from two elements." Similarly, line 51 of the discussion should also be expanded as a paragraph.   

Comments on the Quality of English Language

Some grammatical edits are needed throughout. 

Author Response

Dear editor

Firstly, I would like to thank the reviewers for their comments. They contributed greatly to our group's reflections, as well as to our academic growth.

In our opinion, all of them were relevant and, therefore, addressed.

In order to better guide your evaluation, please find below our response and indication of the location of the corrections.

Throughout the new version, the corrections are highlighted in green.

QUESTION 1

It would be helpful to provide a little more analysis of the role of public policy in influencing public choices. What are some examples that have worked (e.g., anti-smoking campaigns) and when have they not worked? What factors stood out most in successful campaigns? Some minor grammatical errors to consider include:

ANSWER: Two paragraphs have been implemented, located in lines 134-147.

QUESTION 2

Incomplete sentence in line 58: “Furthermore, as the authors point out.”

ANSWER: The adjustments have been made.

QUESTION 3

Define or write out the acronym AFL in line 67, as it is the first time it has been used.

ANSWER: The term “leisure physical activity” has been inserted in full, followed by its abbreviation (AFL). Subsequently, only the abbreviated term has been used throughout the text.

Materials and methods:  Good consideration of the process used for the project.  It would be useful to include more details in the analysis section.  For example, what was learned during the various stages?  Were there any changes in decision-making along the way?  What insights were gained at each stage?

ANSWER: To meet the demand, a paragraph was inserted in the methodology – 198-202.

Results: The use of tables to summarize the results and changes in universities' approaches over the period was a useful presentation of the data. Did the researchers find any significant differences across the periods that stood out as impactful for change in student practices? The 2023–2026 period had the highest number of reported activities. Do they seem to be on track to achieve these goals, considering that the period has not yet been completed?

ANSWER: A paragraph has been included to address the request—page 10, lines 4–6.

Discussion: The discussion section would benefit from further expansion. The authors have outlined the main areas identified through their results and can now elaborate on them. For example, expand the following paragraph, as it is only one sentence:

(line 34) “The absence of explicit objectives for health promotion until 2019 and the implicit attention to this issue from 2020 onwards lead us to analyze them based on two elements.” Similarly, line 51 of the discussion should also be expanded into a paragraph.

ANSWER: The discussions have been expanded to address the request and are highlighted in green.

Reviewer 2 Report

Comments and Suggestions for Authors

Dear Author,

If is possible, in the table 2 add the language in English.

If is possible in the discussion add more recent studies developed by researchers from the universities in table 1 that promote Physical Leisure Activities and Health to the students.

Author Response

Dear editor

Firstly, I would like to thank the reviewers for their comments. They contributed greatly to our group's reflections, as well as to our academic growth.

In our opinion, all of them were relevant and, therefore, addressed.

In order to better guide your evaluation, please find below our response and indication of the location of the corrections.

Throughout the new version, the corrections are highlighted in green.

QUESTION 1

It would be helpful to provide a little more analysis of the role of public policy in influencing public choices. What are some examples that have worked (e.g., anti-smoking campaigns) and when have they not worked? What factors stood out most in successful campaigns? Some minor grammatical errors to consider include:

ANSWER: Two paragraphs have been implemented, located in lines 134-147.

QUESTION 2

Incomplete sentence in line 58: “Furthermore, as the authors point out.”

ANSWER: The adjustments have been made.

QUESTION 3

Define or write out the acronym AFL in line 67, as it is the first time it has been used.

ANSWER: The term “leisure physical activity” has been inserted in full, followed by its abbreviation (AFL). Subsequently, only the abbreviated term has been used throughout the text.

Materials and methods:  Good consideration of the process used for the project.  It would be useful to include more details in the analysis section.  For example, what was learned during the various stages?  Were there any changes in decision-making along the way?  What insights were gained at each stage?

ANSWER: To meet the demand, a paragraph was inserted in the methodology – 198-202.

Results: The use of tables to summarize the results and changes in universities' approaches over the period was a useful presentation of the data. Did the researchers find any significant differences across the periods that stood out as impactful for change in student practices? The 2023–2026 period had the highest number of reported activities. Do they seem to be on track to achieve these goals, considering that the period has not yet been completed?

ANSWER: A paragraph has been included to address the request—page 10, lines 4–6.

Discussion: The discussion section would benefit from further expansion. The authors have outlined the main areas identified through their results and can now elaborate on them. For example, expand the following paragraph, as it is only one sentence:

(line 34) “The absence of explicit objectives for health promotion until 2019 and the implicit attention to this issue from 2020 onwards lead us to analyze them based on two elements.” Similarly, line 51 of the discussion should also be expanded into a paragraph.

ANSWER: The discussions have been expanded to address the request and are highlighted in green.

EVALUATOR 2

If possible, add the English language to Table 2.

ANSWER: Corrected

If possible, in the discussion, add more recent studies developed by researchers from the universities in Table 1 that promote Physical Leisure and Health Activities for students.

ANSWER: Three paragraphs were added starting on page 11, line 100.